# PTQTP: POST-TRAINING QUANTIZATION TO TRIT-PLANES FOR LARGE LANGUAGE MODELS

## ABSTRACT

Post-training quantization (PTQ) of large language models (LLMs) to extremely low bit-widths remains challenging due to the fundamental trade-off between computational efficiency and model expressiveness. While existing ultra-low-bit PTQ methods rely on binary approximations or complex compensation mechanisms, they suffer from either limited representational capacity or computational overhead that undermines their efficiency gains. We introduce **PTQ** to **T**rit-**P**lanes (PTQTP), the first ternary-weight PTQ framework that decomposes weight matrices into structured ternary $\{-1, 0, 1\}$ trit-planes using 2×1.58-bit representation. PTQTP achieves multiplication-free inference, identical to 1-bit quantization, while maintaining superior expressiveness through its novel structured decomposition. Our approach provides: (1) a theoretically grounded progressive approximation algorithm ensuring global weight consistency; (2) model-agnostic deployment across diverse modern LLMs without architectural modifications; and (3) uniform ternary operations that eliminate the need for mixed-precision or compensation schemes. Comprehensive experiments across LLaMA3.x and Qwen3 model families (0.6B-70B parameters) demonstrate that PTQTP significantly outperforms existing low-bit PTQ methods, achieving 82.4% mathematical reasoning retention versus 0% for competing approaches. PTQTP approaches and sometimes surpasses 1.58-bit quantization-aware training performance while requiring only single-hour quantization compared to 10-14 GPU days for training-based methods. These results establish PTQTP as a practical solution for efficient LLM deployment in resource-constrained environments.

## 1 INTRODUCTION

The unprecedented success of large language models (LLMs) has revolutionized natural language processing, but their deployment is hindered by exorbitant computational and memory demands (Brown et al., 2020). Models such as DeepSeek-R1-671B (Guo et al., 2025), with hundreds of billions of parameters, require specialized hardware and massive energy consumption, limiting accessibility on edge devices and raising environmental concerns (Patterson et al., 2021). To address this, extreme low-bit quantization, viz. binary (1-bit) or ternary (1.58-bit), has emerged as a viable approach, facilitating multiplication-free operations that align with efficient hardware implementation.

Post-training quantization (PTQ) offers a practical pathway for compressing pretrained LLMs without retraining, with recent advancements like GPTQ (Frantar et al., 2022) and AWQ (Lin et al., 2024) achieving effective 4-bit quantization. However, pushing PTQ to lower bit-widths, say, 1 to 2 bits, remains a significant challenge. Current approaches bifurcate into two streams: Quantization-aware training (QAT), which requires costly retraining (e.g., BitNet (Wang et al., 2023; Ma et al., 2024) for pretraining binary/ternary models), and binary PTQ which achieves 1-bit Quantization through unstructured weight categorization but sacrifices representational capacity (e.g., BiLLM (Huang et al., 2024) and ARB-LLM (Li et al., 2025)). Structured ternary PTQ, offering higher expressivity than binary while avoiding QAT's pretraining overhead, remains underexplored (if not unexplored).

We introduce PTQTP (**P**ost-**T**raining **Q**uantization to **T**rit-**P**lanes), a novel structured PTQ method that bridges binary efficiency and ternary expressiveness. PTQTP decomposes the full-precision weight matrices into trit-planes ($\{-1, 0, 1\}$) with scaling coefficients. By combining the computational efficiency of binary quantization with the expressiveness of ternary representations, PTQTP offers a

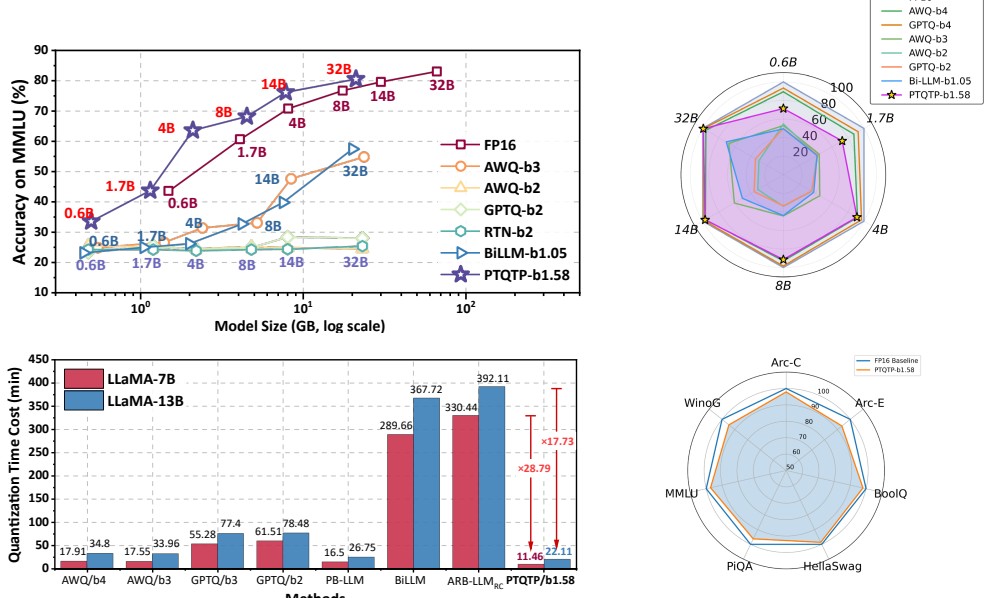

Figure 1: Qwen3 performance evaluation. (a) PTQTP outperforms existing low-bit quantization methods (1/2/3-bit) while maintaining equivalent model size. (b) Quantization runtime comparison showing PTQTP achieves 17.73×-28.79× speedup over ARB-LLM$_{RC}$ and 1.53×-1.57× over AWQ. (c) PTQTP performance approaches both 4-bit quantization and FP16 baselines across model scales. (d) Minimal precision degradation observed across all benchmarks when applying PTQTP to Qwen32B.

practical and uniform solution for deploying powerful LLMs on resource-constrained platforms. Our contributions are as follows:

- To our knowledge, PTQTP is the first 1.58-bit PTQ framework, addressing a critical gap in the literature. It decomposes weight matrices into trit-planes $\{-1, 0, 1\}$ without retraining or fine-tuning, enabling hardware-efficient, multiplication-free operations while more effectively capturing weight distributions than binary PTQ.

- PTQTP offers a robust, model-agnostic solution for extreme low-bit PTQ. By eliminating the need for architecture-specific adjustments, it supports seamless deployment on quantization-sensitive models (e.g., LLaMA3.x (Grattafiori et al., 2024), Qwen3 (QwenTeam, 2025)), while consistently preserving performance and surpassing state-of-the-art architecture-dependent approaches in scalability.

- Extensive experiments demonstrate that PTQTP consistently outperforms state-of-the-art low-bit methods, surpassing most 1–3 bit PTQ approaches on language benchmarks while enabling faster quantization. Remarkably, it rivals or even exceeds 1.58-bit QAT—despite requiring over $10^4\times$ fewer GPU hours—without any retraining or post-PTQ fine-tuning, underscoring its efficiency and generalizability across advanced models.

By addressing the core challenges of expressiveness, stability, scalability, and hardware efficiency, PTQTP sets a new yardstick for compressing high-performance LLMs for resource-constrained platforms. Our work demonstrates that structured ternary quantization strikes a sweet spot between computational simplicity with representational power, as shown in Fig.1, opening new frontiers for efficient inference in real-world applications.

## 2 RELATED WORK

**PTQ for LLMs.** PTQ has emerged as a promising approach for compressing LLMs without the computational overhead of retraining. GPTQ (Frantar et al., 2022) pioneered efficient quantization through layer-wise optimization, achieving 4-bit quantization while maintaining model performance. AWQ (Lin et al., 2024) further improved this by incorporating activation-aware scaling, demonstrating that the consideration of activations during quantization leads to better performance. Recent work has pushed towards even lower bit-width quantization. PBLLM (Shang et al., 2023) and BiLLM (Huang et al., 2024) achieved 1-bit quantization by identifying and preserving structurally salient weights while applying different quantization strategies to different weight groups. ARB-LLM (Li et al., 2025) further refined this approach through alternating optimization of binary weight groups. However, these methods rely on complex, unstructured weight categorizations that complicate hardware implementation.

**Ternary Quantization** The transition from binary to ternary quantization represents a leap of model expressiveness while still preserving multiplier-less arithmetic. BitNet's (Wang et al., 2023) extension to 1.58-bit (ternary) weights (Ma et al., 2024) demonstrated superior performance compared to binary models, suggesting the value of the additional state in the ternary representation. There are also some other trit-plane-based works on image-compression domain (Lee et al., 2022; Jeon et al., 2023).

**Hardware Efficiency.** A key advantage of binary and ternary quantization lies in their suitability for FPGA or ASIC deployment (e.g., FlightLLM (Zeng et al., 2024), DFX (Hong et al., 2023)), where multiplier-less feature–weight products can be efficiently implemented in hardware. Specifically, binary operations can be realized with XNOR gates and bit-counting, while ternary operations leverage simple adders, preserving computational efficiency while offering greater expressivity.

**Existing Problems.** Existing ternary quantization methods mainly (if not all) rely on QAT approaches (such as BitNet), requiring extensive pretraining and substantial computational resources. In addition, previous methods use unstructured weight quantization to compensate for the degenerate representation ability. Unlike these works that optimize progressive reconstruction via retrained context models, PTQTP is the first post-training, structured and model-agnostic framework that decomposes LLM weights into uniform trit-planes with adaptive ridge-scaled coefficients to yield 1.58-bit quantization without retraining.

## 3 METHODOLOGY

The core idea of PTQTP is to progressively minimize the difference between the quantized matrix and the original weights. Therefore, we first focus on finding the optimal ternary patterns $T^{(k)} \in \{-1, 0, 1\}^{n \times d}$ and scaling coefficients $\alpha^{(k)} \in \mathbb{R}^n$, trying to determine the best combination of values from the set $\{-1, 0, 1\}$ for each element of the $k$th trit-plane such that the approximation of the original weight matrix $W$ is as accurate as possible. The use of trit-planes allows multiplication-free operations, which can significantly reduce the compute cost. Then we use an adaptive regularization parameter to define the stability for the auto-adjustment process. This helps in ensuring that the optimization process is stable and converges to a reasonable solution. The general process of PTQTP is illustrated in Fig.2.

### 3.1 DIRECT TRIT-PLANE APPROXIMATION

Consider a pretrained LLM with a weight matrix $W \in \mathbb{R}^{n \times d}$. Our objective is to decompose it into two trit-planes with optimal scaling coefficients: $W \approx \hat{W} = \sum_{k=1}^{2} \text{diag}(\alpha^{(k)}) T^{(k)}$. Such a two-plane approach is akin to the residual/error compensation bit-plane in (Huang et al., 2024; Li et al., 2025), though PTQTP treats the two planes unanimously in the approximation process.

**Row-wise Decomposition.** We adopt a row-wise decomposition approach to find the optimal ternary row and scaling. We use $T_i^{(k)}$ to denote the $i$th row of the $k$th trit-plane. First we fix $T_i^{(k)}$ and optimize scaling coefficients $\alpha_i^{(k)}$ without any bias terms. To solve for optimal scaling coefficients $\alpha_i^{(k)}$, we use the adaptive ridge regression, i.e., linear regression with a regularization term to the least-squares function. We first define the local basis matrix $S_i = \left[ (T_i^{(1)})^T \ (T_i^{(2)})^T \right] \in \{-1, 0, 1\}^{d \times 2}$, and

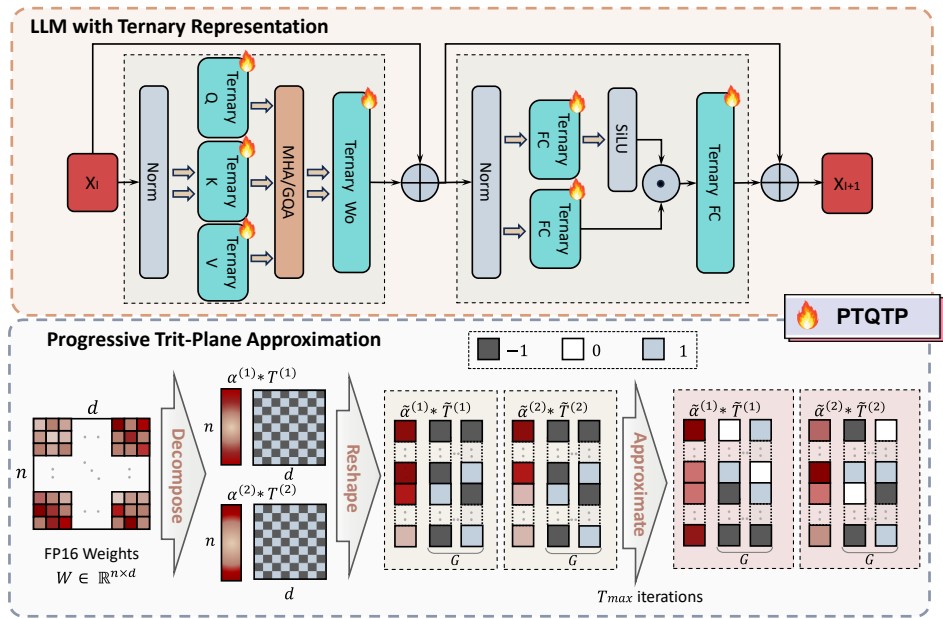

Figure 2: PTQTP workflow overview: (top) Linear layer transformation pathway for ternary quantization in LLaMA architecture; (bottom) Group-wise progressive trit-plane approximation process, where $G$ represents group size and $T_{max}$ indicates maximum iteration count.

$$A_i = (S_i)^T S_i + \lambda_i I_2 \in \mathbb{R}^{2 \times 2}, \quad b_i = (S_i)^T W_i^T \in \mathbb{R}^2, \tag{1}$$

Here $\lambda_i \in \mathbb{R}$ is a regularization parameter to improve numerical stability, $I_2$ is the $2 \times 2$ identity matrix. By constructing the local basis matrix row by row, we can find a closed-form solution $\alpha_i(\in \mathbb{R}^2) := [\alpha_i^{(1)} \alpha_i^{(2)}]^T = A_i^{-1} b_i$ for each $i$ independently. However, due to the discrete nature of trit-planes, the method may converge to different minima depending on the regularization parameter $\lambda_i$, which is crucial for the regression performance: if it is too small, the regularization effect may be negligible and the solution may be unstable. If it is too large, the coefficients $\alpha_i$ can be overly diminished, leading to a poor approximation of the original weight matrix.

### 3.2 Fine-Grained Optimization

**Progressive and Adaptive Regularization.** To achieve a robust decomposition process, we adaptively regularize the approximation and further optimize it through progressive optimization. First, we estimate the condition number of the $2 \times 2$ system:

$$\kappa_{i,approx} = \|A_i\|_F \cdot \|A_i^{-1}\|_F \tag{2}$$

This measure guides dynamic updates to $\lambda_i$ with the constraint $\lambda_{i,new} \leq \lambda_{\max}(:= 1.0)$ :

$$\lambda_{i,new} = \begin{cases} \lambda_i & , \quad \kappa_{i,approx} < 10^{12} \\ \lambda_i \sqrt{\kappa_{i,approx}/10^{12}} & , \quad \text{otherwise} \end{cases} \tag{3}$$

This adaptation mitigates under-regularization (leading to singularity and unstable solutions) and over-regularization (causing excessive coefficient shrinkage), ensuring robustness across weight blocks with varying numerical conditions. Furthermore, the optimal scaling coefficients $\alpha_i$ are then found by solving the following optimization problem:

$$\alpha_i = \arg \min_{\theta_i \in \mathbb{R}^2} \|W_i - S_i \theta_i\|_F^2 + \lambda_i \|\theta_i\|_F^2 \tag{4}$$

Where $\theta_i$ is a crucial variable that represents the intermediate solution to the scaling coefficients in the process of updating $\alpha_i$. In addition, $\lambda_i \|\theta_i\|_F^2$ is the regularization term that penalizes large values of

coefficients $\theta_i$. The Frobenius norm term represents the squared error between the linear combination of the $W_i$ and $\hat{W}_i$, where $S_i$ incorporates the current iteration's trit-plane values. We adapt $\lambda_i$ using Eq. equation 3, aligning regularization strength with local weight properties. Specifically, we perform the optimization steps $\leq T_{max}$. Each update step is designed to not increase the Frobenius norm $\|W - \hat{W}\|_F^2$. By ensuring that this norm monotonically decreases, we can guarantee that the algorithm will converge to a local minimum. Therefore, we refine trit-plane elements through local exhaustive search, updating $T_{i,j}^{(k)}$ to the value in $c_m \in \{-1, 0, 1\}$ that minimizes squared error:

---

**Algorithm 1** Fine-Grained Group-wise PTQTP

---

**Require:** Weight matrix $W$, group size $G$, max iterations $T_{\max}$, tolerance $\epsilon$
**Ensure:** Quantized weight approximation $\hat{W}$
1: Divide $W$ into groups $\tilde{W}_i \in \mathbb{R}^G$          $\triangleright$ Each row $i$ contains $G$ weights
2: Initialize $\tilde{T}_{i,(0)}^{(k)} \leftarrow \text{sign}(\tilde{W}_i)$, $\tilde{\alpha}_{i,(0)}^{(k)} \leftarrow [1, 1]$          $\triangleright$ Initial trit-planes
3: **for** $t \leftarrow 1$ to $T_{\max}$ **do**
4:      **for** each row $i$ **do** row-wise decomposition
5:          $\tilde{S}_i \leftarrow [(\tilde{T}_{i,(t-1)}^{(1)})^T \quad (\tilde{T}_{i,(t-1)}^{(2)})^T]$          $\triangleright$ Basis matrix construction
6:          $\tilde{\alpha}_{i,(t)}^{(k)} \leftarrow \text{RidgeRegression}(\tilde{S}_i, \tilde{W}_i, \lambda_i)$
7:      **end for**
8:      **for** each row $i$ **do**
9:          $\tilde{T}_{i,(t)}^{(1)}, \tilde{T}_{i,(t)}^{(2)} \leftarrow \arg\min_{c_m^{(1)}, c_m^{(2)}} \|\tilde{W}_i - \tilde{\alpha}_{i,(t)}^{(1)} c_m^{(1)} - \tilde{\alpha}_{i,(t)}^{(2)} c_m^{(2)}\|_F^2$
10:      **end for**
11:      **if** $\max_i \|\alpha_{i,(t)}^{(k)} - \alpha_{i,(t-1)}^{(k)}\|_F < \epsilon$ **then break**
12:      **end if**
13: **end for**

---

$$T_{i,j}^{(k)} = \arg\min_{c_m^{(k)} \in \{-1,0,1\}} \left( W_{ij} - \sum_{k=1}^2 \alpha_i^{(k)} c_m^{(k)} \right)^2 \tag{5}$$

Unlike the unstructured mask used for salient weights search in (Huang et al., 2024; Li et al., 2025), PTQTP achieves $\mathbb{E}[W_{ij}] \approx \alpha_i^T \cdot \mathbb{E}\left[[T_{ij}^{(1)} \ T_{ij}^{(2)}]^T\right]$ and is bias-free and mask-free. This avoids extra bias parameter storage and its uniform model architecture preserves hardware-friendly design and operations. Details of the progressive and adaptive regularization algorithm implementation are in Appendix B.

**Group-wise Approximation.** Following the previous works (Frantar et al., 2022; Lin et al., 2024), we further introduce group-wise (aka block-wise) processing in PTQTP, namely, by reshaping $W$ from $n \times d$ to $\frac{nd}{G} \times G$, or equivalently grouping into $G$ columns. Specifically, we set $G = 128$ similar to other works for balanced performance and overhead, and we denote such group-wise operation with the tilde $(\tilde{\circ})$ notation, such that Eq. equation 1 becomes

$$\tilde{A}_i = \tilde{S}_i^T \tilde{S}_i + \lambda_i I_2; \quad \tilde{b}_i = \tilde{S}_i^T \tilde{W}_i^T; \quad \tilde{\alpha}_i = \tilde{A}_i^{-1} \tilde{b}_i \tag{6}$$

Such grouping of columns generally reshapes $W$ into a taller $\tilde{W}$, alongside with lengthened $\tilde{\alpha}^{(k)}$'s. Nonetheless, the increase in the latter incurs negligible parameter overhead compared to the model size, which is far outweighed by an improved approximation and performance. Algorithm 1 summarizes this group-wise PTQTP.

For each linear layer, we first decompose the FP16 weights into trit-planes for direct approximation. Then we apply the progressive group-wise optimization and adaptive regulation to save the optimal parameters automatically. This approach automatically searches for appropriate quantization parameters in the algorithm, making the model parameter distribution more suitable for quantization. We also analyze the convergence of PTQTP in Appendix C. *Experiment results demonstrate that this approximation process always converges within 50 iterations.*

Table 1: Perplexity ($\downarrow$) comparison across SOTA model families on WikiText2 dataset (group size=128). N/A: Model size variant not available; OOM: Out of memory on single A100 80GB GPU. $\dagger$LLaMA-3.1 (8B/70B), LLaMA-3.2 (1B/3B), Qwen2.5 (0.5B/1.5B/3B/7B/32B). *Qwen3 (0.6B/1.7B/4B/8B/32B). **Bold**: best result; underlined: second-best.

| Model | Method | # Bits | 0.5/0.6B* | 1$^\dagger$/1.5/1.7B* | 3$^\dagger$/4B* | 7/8B*$^\dagger$ | 32/70B$^\dagger$ |
|---|---|---|---|---|---|---|---|
| | FP16 | 16.00 | N/A | 9.75 | 7.80 | 6.23 | 2.81 |
| | AWQ | 3.00 | N/A | 34.81 | 13.22 | 10.80 | 14.55 |
| | AWQ | 2.00 | N/A | 1.64E5 | 1.11E5 | 7.98E5 | 4.52E4 |
| | GPTQ | 3.00 | N/A | 18.99 | 16.34 | 9.12 | 8.14 |
| LLaMA 3.1/3.2 | GPTQ | 2.00 | N/A | 4.97E3 | 2.42E3 | 7.17E2 | 18.79 |
| | BiLLM | 1.06 | N/A | 1.99E3 | 1.24E2 | 87.25 | 1.82E2 |
| | ARB-LLM$_{RC}$ | 1.06 | N/A | 1.25E2 | 38.67 | 45.49 | OOM |
| | PTQTP | 1.58 | N/A | **17.15** | **10.24** | **8.53** | **7.76** |
| | FP16 | 16.00 | 9.28 | 13.08 | 8.03 | 6.85 | 5.02 |
| | AWQ | 3.00 | 51.30 | 22.48 | 4.57E3 | 12.12 | 6.49 |
| | AWQ | 2.00 | 4.36E5 | 1.18E6 | 5.84E5 | 5.26E5 | 1.16E3 |
| | GPTQ | 3.00 | **20.65** | 13.14 | 2.34E6 | 3.08E4 | 6.19 |
| Qwen 2.5 | GPTQ | 2.00 | 2.35E3 | 6.77E2 | 1.63E6 | 2.81E4 | 14.30 |
| | BiLLM | 1.06 | 3.03E3 | 8.98E2 | 1.48E2 | 60.72 | 13.13 |
| | ARB-LLM$_{RC}$ | 1.06 | 4.84E2 | 3.10E2 | 41.25 | 17.85 | 9.46 |
| | PTQTP | 1.58 | 31.03 | **12.72** | **13.13** | **8.31** | **6.00** |
| | FP16 | 16.00 | 20.9 | 16.70 | 13.64 | 9.71 | 8.64 |
| | AWQ | 3.00 | 2.20E2 | 84.00 | 91.00 | 27.50 | 19.10 |
| | AWQ | 2.00 | 1.21E7 | 7.52E6 | 1.38E7 | 1.21E7 | NAN |
| | GPTQ | 3.00 | 3.14E4 | 6.17E4 | 1.34E5 | 5.02E5 | 37.10 |
| Qwen 3 | GPTQ | 2.00 | 2.38E4 | 6.55E2 | 5.92E5 | 7.77E4 | 38.40 |
| | BiLLM | 1.06 | 5.87E4 | 7.54E4 | 299.86 | 113.45 | 17.10 |
| | ARB-LLM$_{RC}$ | 1.06 | 8.43E2 | 3.26E2 | 50.30 | 32.26 | 12.16 |
| | PTQTP | 1.58 | **38.02** | **32.46** | **18.25** | **11.8** | **10.06** |

## 4 EXPERIMENTS

### 4.1 SETTINGS

**Quantization Configuration.** We implemented PTQTP in PyTorch using models from Hugging-face (Paszke et al., 2019). All linear layers were quantized with tolerance $\epsilon = 10^{-4}$ and maximum iterations $T_{max} = 50$. For numerical stability, we employed dynamic regularization with $\lambda \in [10^{-8}, 1]$. No task-specific calibration, tuning, or fine-tuning was applied in any experiment. All evaluations were conducted on a single NVIDIA A100 80GB GPU.

**Model Architectures.** We evaluated PTQTP across multiple mainstream LLM families including Qwen3 (QwenTeam, 2025), LLaMA3.x (Grattafiori et al., 2024), Qwen 2.5 (Yang et al., 2024), and LLaMA 1& 2 series (Touvron et al., 2023a;b). To assess cross-domain generalization capabilities, we also tested instruction-tuned variants of these models.

**Baseline Methods.** We benchmarked PTQTP against three categories of methods: (1) Binary PTQ methods—PBLLM (Shang et al., 2023), BiLLM (Huang et al., 2024), ARB-LLM (Li et al., 2025); (2) Popular PTQ Methods—GPTQ (Frantar et al., 2022), AWQ (Lin et al., 2024); and (3) 1.58-bit QAT approaches (Ma et al., 2024; 2025) to demonstrate PTQTP's effectiveness even against training-based methods. For fair comparison with smaller models, we included common 1B-3B LLMs such as SmolLM2 (Allal et al., 2025) and MiniCPM (Hu et al., 2024).

**Evaluation Protocol.** We assessed language modeling capability through perplexity measurements on WikiText-2 (Merity et al., 2016), PTB (Marcus et al., 1994), and C4 (Raffel et al., 2020). To evaluate reasoning abilities, we utilized ARC-Challenge, ARC-Easy (Clark et al., 2018), BoolQ (Clark et al.,

2019), HellaSwag (Zellers et al., 2019), PIQA (Bisk et al., 2020), Winogrande (Sakaguchi et al., 2021), and MMLU (Hendrycks et al., 2021). For comprehensive comparison with SOTA methods, we further evaluated coding and mathematical reasoning performance on LLaMA3.x and Qwen3 models. All evaluations were conducted using the standard benchmarking tool (Gao et al., 2024). Detailed analysis of runtime, computational complexity, memory footprint, and inference efficiency are provided in Appendix A.

## 4.2 MAIN RESULTS

> **TL;DR Key Insights and Findings**
>
> - **Robustness Advantage in Smaller Models.** PTQTP demonstrates superior performance retention over traditional low-bit quantization across SOTA models, with perplexity metrics surpassing 1/2/3-bit schemes and approaching 4-bit levels, especially in smaller models (0.6B-3B) that are typically more sensitive to quantization.
>
> - **Breakthrough Preservation of Mathematical Reasoning at Ultra-Low Bits.** Conventional low-bit quantization (e.g., 3-bit/2-bit GPTQ and BiLLM) causes catastrophic degradation of mathematical reasoning capabilities (0% accuracy on Math-500). In contrast, PTQTP-b1.58 achieves only a decrease of less than 5% from the baseline (FP16), fundamentally challenging the precision-performance tradeoff paradigm.
>
> - **Plug-and-Play Efficiency.** PTQTP enables extreme 1.58-bit quantization with zero retraining, eliminates the need for salient weight protection common in competing schemes, and reduces quantization time by up to 28.79× compared to other low-bit PTQ methods.
>
> - **Cross-Architecture and Task Universality.** Unlike QAT, which requires extensive pretraining and fine-tuning for different model architectures, PTQTP retains fidelity across multi-task benchmarks *without* model-specific adjustments, adapting seamlessly to architectural variations from LLaMA3.x to Qwen3 and scaling from 0.6B to 70B parameters.
>
> - **Structured Quantized Data Formate.** PTQTP employs 1.58x2 trit-planes for quantized weights representation, striking a balance between efficiency and accuracy. Similarly to approaches that retain higher precision for salient channels to preserve information, PTQTP introduces a fully symmetric, low-bit auxiliary plane that not only offers high parallelization potential, but also effectively preserves model performance.

**Perplexity Results on SOTA Model Families.** Table 1 compares PTQTP with baselines across LLaMA3.1/3.2, Qwen 2.5 and Qwen 3 variants (0.6B to 70B). PTQTP performances on additional datasets are in Table 9 of the Appendix. The results demonstrate that PTQTP consistently outperforms existing extreme low-bit (1-3 bit) quantization schemes and approaches or exceeds 4-bit methods across diverse architectures. This robustness is particularly pronounced in SOTA small LLMs (0.6B-3B), which are typically more vulnerable to quantization due to their higher information density from advanced pretraining recipes. The exceptional performance retention of PTQTP establishes it as a robust, generalizable, and efficient quantization solution for both current and future models. Additional analyses on robustness are in Appendix D.

**Comparison of PTQ Methods.** We select Qwen3-14B-instruct as our benchmark baseline, as larger models generally exhibit greater resistance to quantization-induced performance loss, making the 14B size ideal for evaluating extreme quantization effectiveness. Results in Table 2 reveal dramatic disparities in capability retention across PTQ methods. GPTQ variants at 3-bit collapse to near-zero performance on mathematical tasks, while specialized 1.05-bit schemes (PBLLM, BiLLM) show catastrophic failure on Math-500 (0%) and GSM8K (<2%). Even ARB-LLM$_{RC}$-b1.05, despite architectural modifications, achieves only 1.80% and 32.22% on these benchmarks—significantly below baseline. *This evidence confirms that mathematical reasoning is uniquely vulnerable to precision loss in conventional PTQ approaches*, exhibiting 19× greater performance degradation compared to linguistic tasks, with non-negligible impacts (12.7% mean reduction) on general reasoning benchmarks. These findings suggest that architectural modifications alone—such as salient weight protection—cannot effectively preserve mathematical reasoning capabilities at ultra-low precision.

In contrast, PTQTP-b1.58 achieves 82.40% on Math-500 and 85.44% on GSM8K—less than 5% degradation from the baseline. This preservation of mathematical reasoning at ultra-low bit-widths

Table 2: Performance comparison of quantization methods on Qwen3-14B across mathematical reasoning and general benchmarks (accuracy %). **Bold**: best result; underlined: second-best.

| Model | Math-500 | GSM8K | ARC-C | ARC-E | BoolQ | HellaSwag | PIQA | MMLU | WinoG |
|---|---|---|---|---|---|---|---|---|---|
| Baseline-FP16 | 86.60 | 89.39 | 95.25 | 90.83 | 89.45 | 88.09 | 81.07 | 79.38 | 68.90 |
| AWQ-b4 | **85.20** | **89.31** | **94.24** | **91.18** | **89.11** | **87.14** | **79.82** | **78.50** | 67.64 |
| GPTQ-b3 | 0.00 | 2.12 | 7.46 | 10.41 | 7.22 | 19.95 | 10.77 | 21.34 | 26.44 |
| PBLLM-b1.05 | 0.00 | 1.21 | 4.75 | 7.76 | 29.14 | 7.38 | 25.46 | 12.11 | 49.57 |
| BiLLM-b1.05 | 0.00 | 0.83 | 50.17 | 4.76 | 49.39 | 22.15 | 7.40 | 24.40 | 49.96 |
| ARB-LLM$_{RC}$-b1.05 | 1.80 | 32.22 | 76.95 | 82.01 | 80.00 | 50.01 | 69.91 | 54.08 | 50.36 |
| PTQTP-b1.58 | 82.40 | 85.44 | 93.56 | 88.54 | 89.02 | 85.12 | 79.05 | 76.18 | **67.88** |

Table 3: Comparison between PTQTP-quantized models and leading instruction-tuned LLMs (1B-4B parameters) across efficiency metrics and benchmark performance (accuracy %). **Bold**: best result; underlined: second-best. * Results claimed by BitNet-b1.58 paper.

| Model (Params) | Math-500 | GSM8K | ARC-C | ARC-E | BoolQ | HellaSwag | PIQA | MMLU | WinoG |
|---|---|---|---|---|---|---|---|---|---|
| LLaMA3.2 (1B) | 14.40 | 46.47 | 55.93 | 69.84 | 62.20 | 40.12 | 58.16 | 46.81 | 50.04 |
| Qwen2.5 (1.5B) | 56.20 | 64.67 | 71.86 | **89.59** | 79.48 | 61.88 | 75.73 | 61.91 | 55.80 |
| Qwen3 (1.7B) | 72.20 | 74.60 | **82.71** | 80.07 | 78.59 | 57.76 | 67.36 | 60.63 | 50.75 |
| SmolLM2 (1.7B) | 21.00 | 50.19 | 49.15 | 76.01 | 67.25 | 50.41 | 67.57 | 49.73 | 50.04 |
| MiniCPM (2B) | 0.60 | 56.48 | 62.03 | 26.46 | 79.76 | 26.49 | 53.75 | 52.08 | 54.22 |
| BitNet-b1.58 (2B)* | 43.40 | 58.38 | 49.91 | 74.79 | 80.18 | 68.44 | **77.09** | 53.17 | **71.90** |
| LLaMA3.2 (3B) | 45.20 | 77.56 | 80.68 | 88.36 | 80.89 | 63.75 | 71.38 | 62.24 | 53.75 |
| Qwen3-PTQTP (1.7B) | 43.80 | 54.21 | 51.53 | 70.02 | 27.55 | 37.92 | 57.89 | 43.82 | 51.07 |
| Qwen2.5-PTQTP (3B) | 42.80 | 62.47 | 75.93 | 85.54 | 78.38 | 60.87 | 66.32 | 56.80 | 56.27 |
| LLaMA3.2-PTQTP (3B) | 34.20 | 65.81 | 74.92 | 82.89 | 74.34 | 55.61 | 64.42 | 55.08 | 51.85 |
| Qwen3-PTQTP (4B) | **82.60** | **87.79** | 76.95 | 86.60 | **84.13** | **71.02** | 73.18 | **63.65** | 53.67 |

suggests PTQTP effectively decouples numerical precision requirements from model parameterization. Its consistent performance across diverse benchmarks (93.56% ARC-C, 76.18% MMLU) without requiring dynamic bit allocation, salient weight protection, or sensitivity-aware quantization fundamentally challenges the assumed trade-off between low-bit quantization and mathematical reasoning ability.

**Comparison with Baseline and 1.58-bit QAT Language Models.** PTQTP's versatility enables extreme low-bit quantization of advanced models while maintaining near-baseline performance. Unlike QAT methods that require extensive pretraining and fine-tuning, PTQTP applies uniform treatment across model layers without post-quantization adjustments. Table 3 demonstrates PTQTP's minimal performance degradation compared to baselines, and its ability to match or exceed 1.58-bit QAT BitNet schemes (Ma et al., 2024; 2025) at similar model sizes. These results confirm PTQTP's exceptional stability and establish it as a true plug-and-play solution for model-agnostic 1.58-bit quantization. Additional benchmark results are provided in Appendix E.

## 4.3 ABLATION STUDIES

**Progressive Search Iterations.** Fig. 3 identifies a critical 30-iteration threshold that optimally balances perplexity and efficiency across model scales. Within this range, both test models achieve remarkable convergence: LLaMA3.1-8B reduces WikiText2 perplexity by 94%, while LLaMA3.2-3B recovers from catastrophic initialization (6512.0 → 10.24 perplexity) by effectively adapting trit-plane coefficients to weight distributions. Smaller models converge faster due to simpler weight structures, while larger models leverage their expressivity for steeper initial gains. Beyond 30

iterations, perplexity stabilizes while quantization time increases linearly, indicating diminishing returns from additional refinement. This confirms our theoretical guarantee of monotonic convergence and suggests limiting iterations to $\leq 50$ in practical deployments to achieve near-optimal accuracy without computational inefficiency.

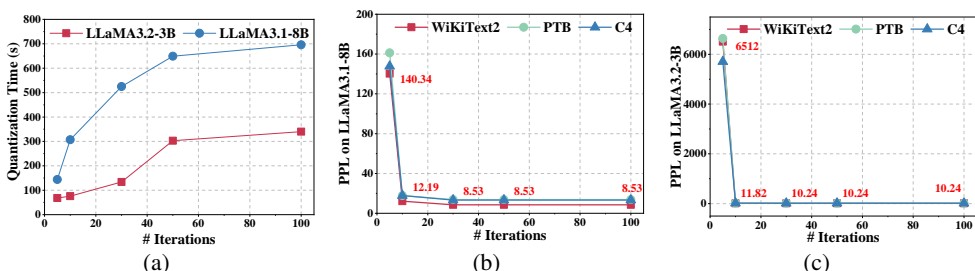

Figure 3: Effect of progressive search iterations on quantization time (left sub-figure) and perplexity (PPL) (middle and right sub-figures) for LLaMA3.1-8B and LLaMA3.2-3B models.

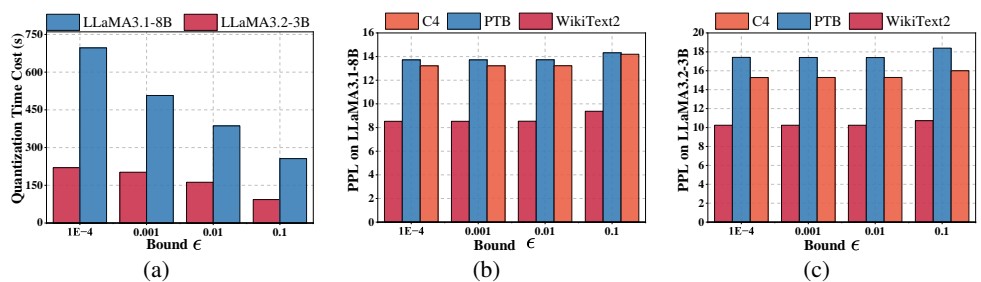

Figure 4: Trade-off between tolerance bounds ($\epsilon$), quantization time (left sub-figure), and model perplexity (PPL) (middle and right sub-figures) for LLaMA3.1-8B and LLaMA3.2-3B architectures.

**Bound of Tolerance** $\epsilon$. Fig. 4 reveals the crucial trade-off between tolerance $\epsilon$ and quantization performance. Tighter $\epsilon$ values improve perplexity but at significant computational cost: LLaMA3.1-8B achieves 9.1% perplexity reduction with a 172% runtime increase, while LLaMA3.2-3B shows 4.5% improvement with 137% longer processing time. The inflection point at $\epsilon = 10^{-2}$ marks where returns diminish substantially—further tightening yields marginal improvements while significantly increasing computation time. Smaller models show better stability at low $\epsilon$, while larger models require stricter tolerance to capture complex weight relationships. Our experiments identify $\epsilon \in [10^{-3}, 10^{-2}]$ as the optimal range for balancing accuracy and efficiency, particularly important for resource-constrained deployment scenarios.

# 5 CONCLUSION

We have introduced PTQTP, a structured PTQ framework that achieves 1.58-bit quantization through systematic trit-plane decomposition, which strikes a balance between the efficiency of binary methods and the representational capacity of higher-precision formats, all without requiring retraining or fine-tuning. Extensive experiments demonstrate that PTQTP outperforms existing low-bit PTQ (1/2/3-bit) methods, and approaches or even surpasses the accuracy of 1.58-bit QAT techniques.

Remarkably, PTQTP preserves mathematical reasoning capabilities that catastrophically degrade in other extreme low-bit quantization schemes, challenging the conventional wisdom that such tasks inherently require higher precision. In addition, PTQTP's model-agnostic design ensures robust deployment across diverse architectures from 1B to 70B models, across most versions of LLaMA and Qwen families.

## LLM USAGE DISCLOSURE

This paper is primarily the work of the human authors, but we also made use of several advanced large language models, including ChatGPT-5, Gemini 2.5, DeepSeek, and Claude-4, as general-purpose assistive tools. They were employed to summarize background literature, provide feedback on research ideas, assist with code development and debugging, support result analysis, and help with formatting and language polishing. We acknowledge the contributions of these LLMs, while fully recognizing their limitations, and take full responsibility for all content presented under our names. We did not include hidden prompt-injection text in the submission, and all external data and code comply with their respective licenses.

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

APPENDIX

# A EFFICIENCY ANALYSIS

> *PTQTP's hardware efficiency is consistent with the binary quantization method. Moreover, the inherent sparsity of trit-plane offers potential opportunities for further quantization to reduce storage requirements and acceleration on application-specific integrated circuit (ASIC).*

## A.1 HARDWARE EFFICIENCY

The PTQTP framework leverages ternary operations $\{-1, 0, 1\}$ to achieve hardware-efficient computations. For a ternary element $c_m \in \{-1, 0, 1\}$ and scaling coefficient $\alpha$, the multiplication operation $\alpha \cdot c_m$ can be implemented as an addition, eliminating floating-point multipliers and replacing them with sign flips, identity mappings, or zeroing operations. This reduces the arithmetic intensity to $\mathcal{O}(1)$ per element in the matrices. The far lower multiplication and the inherent sparsity in PTQTP than other low-bit methods (i.e., 2-bits to 8-bits) and the decomposition into two trit-planes $T^{(1)}, T^{(2)}$, allowing parallelized additive superposition: $\hat{W} = \text{diag}(\alpha^{(1)})T^{(1)} + \text{diag}(\alpha^{(2)})T^{(2)}$, where each plane can be processed independently on parallelized architectures, exploiting data-level parallelism and minimizing latency in inference acceleration. This bandwidth reduction is critical for latency-bound applications, as memory access often dominates inference time in modern hardware architectures, though there are two parallel trit-planes to represent the original weights.

## A.2 COMPUTATION AND TIME COMPLEXITY OF QUANTIZATION

**Total Computational Complexity:** For row $i$, the normal equation matrix $A_i = (S_i)^T S_i + \lambda_i I_2$ is $2 \times 2$, with inverse computed in constant time $\mathcal{O}(1)$ using the adjugate formula:

$$A_i^{-1} = \frac{1}{\det(A_i)} \begin{bmatrix} A_i(2,2) & -A_i(1,2) \\ -A_i(2,1) & A_i(1,1) \end{bmatrix} \tag{7}$$

Where $\det(A_i) = A_i(1,1)A_i(2,2) - A_i(1,2)^2$. The vector $b_i = S_i^T W_i^T$ requires $2d$ operations (dot products for each column of $S_i$), leading to $\mathcal{O}(d)$ per row. Adding up to $n$ rows, this step is $\mathcal{O}(nd)$ per iteration. In addition, for each element $(i, j)$ in $W$, evaluating $(T^{(1)}, T^{(2)}) \in \{-1, 0, 1\}^2$ involves 9 arithmetic operations per element, resulting in $\mathcal{O}(1)$ complexity per element. For $d$ elements per row, this is $\mathcal{O}(nd)$ in $n$ rows per iteration. Therefore, with $T_{\max}$ iterations, the total computational complexity is $\mathcal{O}(T_{\max} \cdot nd)$.

**Total Time Complexity**: For each iteration, the time complexity can be presented as follows:

$$\underbrace{\mathcal{O}(nd)}_{\substack{\text{Ridge Regression} \\ \text{solve for } \alpha}} + \underbrace{\mathcal{O}(nd)}_{\substack{\text{Trit-Plane} \\ \text{element-wise search}}} = \mathcal{O}(nd); \underbrace{\mathcal{O}(n)}_{\substack{\text{Convergence} \\ \text{Check}}} \ll \mathcal{O}(nd) \tag{8}$$

With empirical convergence in $T_{\max} \leq 50$ iterations (Appendix C), the total time complexity is $\mathcal{O}(T_{\max} \cdot nd)$, exhibiting linear scalability with the dimension of weight matrix $n, d$, critical for deployment on LLMs with billions of parameters. $\mathcal{O}(T_{\max} \cdot nd)$ complexity is optimal for low-bit quantization when considering both approximation quality and computational efficiency. Compared to PTQ methods like GPTQ (Frantar et al., 2022) and AWQ (Lin et al., 2024), *PTQTP achieves lower per-iteration complexity due to its closed-form solutions and fixed-dimensional ridge regression.*

For example, GPTQ's iterative Hessian-based optimization has a per-iteration complexity of $\mathcal{O}(nd^2)$, which becomes infeasible for large $d$. The experiment results in Fig. 3 show that PTQTP can stably converges in $\leq 50$ iterations with different model structures, resulting in 1.2-28.79 $\times$ speedup during quantization on modern GPUs, which are shown in Fig. 1(b).

## A.3 SPACE COMPLEXITY AND MEMORY SAVING

**Space Complexity.** A matrix $W \in \mathbb{R}^{n \times d}$ in FP16 format requires $2nd$ bytes (2 bytes per element). However, each ternary element $c_m \in \{-1, 0, 1\}$ can be encoded with 2 bits (since $3 \leq 2^2$), thus two

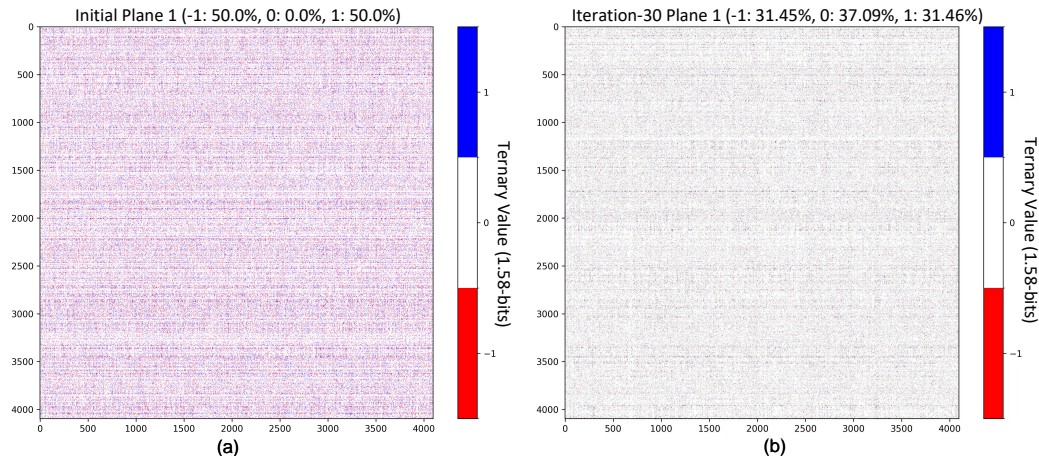

Figure 5: The single trit-plane update process during the optimization iterations on LLaMA3.1-8B.

Table 4: Memory footprint overhead comparison between PTQTP and binary methods including: PB-LLM, BiLLM, ARB-LLM.

| LLaMA - 7B | | | | LLaMA - 13B | | |
|---|---|---|---|---|---|---|
| Method | Group | Memory | | Method | Group | Memory |
| FP16 | - | 13.48 GB | | FP16 | - | 26.03 GB |
| PB-LLM | - | 2.91 GB | | PB-LLM | - | 5.33 GB |
| BiLLM | - | 2.93 GB | | BiLLM | - | 5.36 GB |
| ARB-LLM$_{RC}$ | ✗ | 2.63 GB | | ARB-LLM$_{RC}$ | ✗ | 4.77 GB |
| ARB-LLM$_{RC}$ | ✓ | 2.83 GB | | ARB-LLM$_{RC}$ | ✓ | 5.17 GB |
| PTQTP | ✗ | 3.51 GB | | PTQTP | ✗ | 6.53 GB |
| PTQTP | ✓ | 3.69 GB | | PTQTP | ✓ | 6.89 GB |

trit-planes stored by $nd/2$ bytes. Moreover, two vectors $\alpha^{(1)}, \alpha^{(2)} \in \mathbb{R}^n$ in FP16 require $4n$ bytes. Therefore, the total storage complexity can be presented as $\mathcal{O}(nd)$, achieving a compression ratio of $4\times$ for the trit-plane relative to FP16.

For example, for a typical LLM layer with $n = 1024$ and $d = 4096$, PTQTP reduces storage from 8MB to 1.004MB (i.e., 0.5 MB for trit-planes and 0.504 MB for scaling coefficients), illustrating a $7.96\times$ compression ratio. This reduction enables deploying LLMs on devices with limited memory, such as edge GPUs with 8 GB RAM or mobile devices with 4 GB RAM. The optimized trit-planes examples during the iteration progress are illustrated in Fig. 5.

**Memory Saving.** For $W \in \mathbb{R}^{n \times d}$, group size $k$, the memory of $\hat{W}$ after standard quantization of bits $m$ is

$$\mathcal{M} = \underbrace{n \times d \times m}_{B} + \underbrace{[d/k]}_{\text{multiple groups}} \times \underbrace{n \times 16}_{\text{row-wise FP16 } \alpha} \tag{9}$$

The memory required by BiLLM can be formulated as follows, where $c$ is the number of salient columns for $W$.

$$\mathcal{M}_{\text{BiLLM}} = \underbrace{2 \times n \times c + [d/k] \times 3n \times 16}_{\text{second-order binarization}} + \underbrace{n \times (d - c) + \overbrace{[d/k] \times 2n \times 16 \times 2}^{\text{2 groups}}}_{\text{first-order binarization}}$$

$$+ \underbrace{n \times d}_{\text{group bitmap}} + \underbrace{d}_{\text{salient column bitmap}} \tag{10}$$

Previous work (Li et al., 2025) derived the total memory occupation ARB-RC and ARB-RC + CGB (grouped column bitmap), which can be formulated as:

$$\mathcal{M}_{\text{ARB-RC}} = \underbrace{2 \times n \times c + ([d/k]] \times 2n + 2c) \times 16}_{\text{second-order binarization}}$$

$$+ \underbrace{n \times (d - c) + (\overbrace{[d/k] \times n + (d - c)}^{\text{2 groups}}) \times 16 \times 2}_{\text{first-order binarization}}$$

$$+ \underbrace{n \times d}_{\text{group bitmap}} + \underbrace{d}_{\text{salient column bitmap}} \tag{11}$$

$$\mathcal{M}_{\text{ARB-RC+CGB}} = \underbrace{2 \times n \times c + ([d/k] \times 2n + 2c) \times 16 \times 2}_{\text{second-order binarization}}$$

$$+ \underbrace{n \times (d - c) + (\overbrace{[d/k] \times n + (d - c)}^{\text{2 groups}}) \times 16 \times 2}_{\text{first-order binarization}}$$

$$+ \underbrace{n \times d}_{\text{group bitmap}} + \underbrace{d}_{\text{salient column bitmap}} \tag{12}$$

In PTQTP, each trit-plane containing 3 states has to be stored as a 2bit datatype due to the hardware constraint. The total memory of PTQTP is

$$\mathcal{M}_{\text{PTQTP}} = \underbrace{2 \times n \times d \times 2}_{\text{2 Trit-Plane}} + \underbrace{[d/k] \times 2n \times 16}_{\text{row-wise FP16 } \alpha} \tag{13}$$

Table 4 compares the estimated memory demand of PTQTP with other extreme quantization methods, derived from the above formulas. The proposed PTQTP slightly increased the memory consumption to other binary quantization approaches. *This is a trade-off between storage and representational capacity.* However, methods such as BiLLM and ARB-LLM$_{\text{RC}}$ explicitly divide columns into first-order and second-order groups based on their saliency (as shown in Eqs.10, 11, 12), assigning more bit planes and FP16 parameters to the more salient second-order columns. As a result, although PTQTP uses trit-planes to represent quantized weights, it does not incur significant memory overhead compared to binary-based methods.

### A.4 INFERENCE SPEED

We designed a GPU kernel for efficient inference based on PTQTP. Its performance was evaluated on an NVIDIA RTX 4090 GPU, using different layers from LLaMA2 as benchmarks.

**Linear Layer.** We evaluated the inference latency of the proposed PTQTP kernel on the gate projection layers of LLaMA2 of varying sizes, compared to the GPTQ, AWQ, and AQLM kernels. The tests covered both the prefill and decode stages across a range of sequence lengths, with results summarized in the Table 5.

During the decode stage, PTQTP achieves up to 1.14× speedup. In the prefill stage, all kernels tend to incur higher latency, but PTQTP exhibits significantly smaller latency increase compared to AWQ and AQLM. Overall, PTQTP delivers inference speed second only to GPTQ 4-bit, while maintaining better performance.

**Attention Layer.** For LlamaAttention, we designed a dedicated PTQTP kernel, and evaluated its inference latency during the decode stage across models of different scales. As shown in the Table 6, the proposed kernel achieves up to 1.16× speedup on the 70B model.

## B DETAILS OF PROGRESSIVE AND ADAPTIVE REGULARIZATION

The proposed progressive and adaptive algorithm (Algorithm 2) iteratively refines both the continuous scaling coefficients $\alpha$ and the discrete trit-planes $T^{(1)}, T^{(2)}$ to minimize the approximation error

Table 5: The inference latency (ms) of Gate_proj on Nvidia RTX 4090 GPU.

| LLaMA2 | Sequence len. | FP16 | GPTQ/4bit | AWQ/4bit | AQLM/2x2bit | TPQTP/1.58bit |
|--------|---------------|------|-----------|----------|-------------|---------------|
| **7B** | 1 | 0.122 | 0.085 | 0.092 | 0.049 | 0.120 |
|        | 128 | 0.164 | 0.159 | 0.325 | 5.656 | 0.208 |
|        | 2048 | 1.153 | 1.693 | 4.890 | 85.425 | 2.094 |
| **13B** | 1 | 0.176 | 0.114 | 0.136 | 0.063 | 0.170 |
|        | 128 | 0.241 | 0.272 | 0.489 | 6.868 | 0.310 |
|        | 2048 | 2.154 | 2.654 | 7.428 | 110.051 | 3.661 |
| **70B** | 1 | 0.519 | 0.274 | 0.380 | 0.217 | 0.454 |
|        | 128 | 0.630 | 0.712 | 1.534 | 25.798 | 0.856 |
|        | 2048 | 5.945 | 8.813 | 23.562 | 413.290 | 13.442 |

Table 6: The inference latency (ms) and speedup of LlamaAttention.

| LLaMA2 | 7B | 13B | 70B |
|--------|-----|------|------|
| FP16 | 0.224 | 0.345 | 0.460 |
| PTQTP/1.58bit | 0.197 | 0.317 | 0.395 |
| Speedup | 1.141× | 1.087× | 1.163× |

of the weight matrix $W \in \mathbb{R}^{n \times d}$. In the beginning, the trit-planes $T^{(k)}$ are initialized using the sign function of $W$, with zero entries replaced by 1 to ensure valid ternary values (later expanded to include 0 through optimization); scaling coefficients $\alpha$ are initialized as uniform vectors $[1, 1]$ replicated across all rows; and the regularization parameter $\lambda$ starts at a small value $10^{-8}$ to promote numerical stability.

In each iteration, the algorithm first updates the continuous scaling coefficients $\alpha$ row by row. If the condition number $\kappa_{i,approx} \geq 10^{12}$, indicating ill-conditioning, the regularization parameter $\lambda_i$ is adaptively increased $\leq \lambda_{max} = 1.0$ to stabilize the solution. This adaptive regularization mitigates the sensitivity to small input perturbations. The scaling coefficients are then solved via closed-form ridge regression, ensuring efficient computation without iterative solvers. Next, the algorithm optimizes the discrete trit-planes. It generates all 9 possible ternary value combinations $\mathcal{C} = \{-1, 0, 1\}^2$ for the pair $(T^{(1)}, T^{(2)})$, and for each matrix element $(i, j)$, computes the approximation error for each combination. The combination $\mathbf{c}_{m^*}$ that minimizes the squared error $e_m$ is selected to update $T_{ij}^{(1)}$ and $T_{ij}^{(2)}$, effectively conducting a local exhaustive search to find the best ternary representation for each weight entry. Finally, convergence is checked by monitoring the Frobenius norm difference between consecutive scaling coefficient updates. If $\|\alpha_{(t)} - \alpha_{(t-1)}\|_F < \epsilon$, the algorithm terminates early, balancing optimization quality with computational efficiency.

## C  CONVERGENCE ANALYSIS

In this section, we present a comprehensive convergence analysis of our progressive optimization scheme. Our goal is to provide rigorous theoretical proofs that establish the monotonic reduction of approximation error and the attainment of local optimality. This analysis not only elucidates the trade-off between computational complexity and model expressiveness, but also offers a theoretical foundation for deploying our method on resource-constrained devices.

### C.1  REGULARIZED INITIALIZATION

We adopt a progressive initialization strategy to kick-start the optimization process. The first step of our initialization focuses on approximating the weight matrix $\tilde{W}_i$ group by group. We start by

**Algorithm 2** Progressive and Adaptive Regularization

**Require:** Weight matrix $W \in \mathbb{R}^{n \times d}$, max iterations $T_{\max}$, tolerance $\epsilon$
**Ensure:** Optimized parameters $\alpha, T^{(1)}, T^{(2)}$
1: **Initialize:**
2: $T_{(0)}^{(k)} \leftarrow \text{sign}(W)$ with $0 \rightarrow 1$ replacement for $k = 1, 2$       ▷ Initial trit-planes
3: $\alpha_{(0)} \leftarrow [1, 1] \otimes \mathbf{1}_n$       ▷ Uniform initial scaling coefficients
4: $\lambda_{(0)} \leftarrow 10^{-8} \cdot \mathbf{1}_n$       ▷ Initial regularization parameters
5: **for** $t = 1$ **to** $T_{\max}$ **do**
6:   **Update Continuous Scaling Coefficients:**
7:   **for** row $i = 1$ **to** $n$ **do**
8:    $S_i \leftarrow [T_{i,(t-1)}^{(1)T} \quad T_{i,(t-1)}^{(2)T}]$       ▷ Local basis matrix
9:    $A_i \leftarrow (S_i)^T S_i + \lambda_i I_2$
10:    $b_i \leftarrow (S_i)^T W_i^T$
11:    Solve $\alpha_{(i,t)} \leftarrow (A_i)^{-1} b_i$       ▷ Ridge regression solution
12:    $\lambda_{i,new} \leftarrow \lambda_i \sqrt{\kappa_{i,approx}/10^{12}}$ when $\kappa_{i,approx} \geq 10^{12}$
13:   **end for**
14:   **Optimize Discrete Trit-Planes:**
15:   $\mathcal{C} \leftarrow \{-1, 0, 1\}^2$       ▷ All possible ternary value pairs
16:   **for** element $(i, j) \in W$ **do**
17:    Evaluate error for all $\mathbf{c}_m \in \mathcal{C}$:
18:    $e_m \leftarrow \left( W_{ij} - \sum_{k=1}^2 \alpha_{i,(t)}^{(k)} c_m^{(k)} \right)^2$
19:    $m^* \leftarrow \arg\min_m e_m$       ▷ Select best ternary pair
20:    $T_{ij,(t)}^{(1)}, T_{ij,(t)}^{(2)} \leftarrow \mathbf{c}_m^*$       ▷ Update trit-plane elements
21:   **end for**
22:   **Convergence Check:**
23:   **if** $\|\alpha_{(t)} - \alpha_{(t-1)}\|_F < \epsilon$ **then**
24:    **break**       ▷ Early termination if converged
25:   **end if**
26: **end for**
27: **Return** $\alpha_{(t)}, \{T_{(t)}^{(1)}, T_{(t)}^{(2)}\}$

initializing the trit-planes $\tilde{T}_{i,(0)}^{(1)}$ $\tilde{T}_{i,(0)}^{(2)}$ using the sign function of $\tilde{W}_i$:

$$\tilde{T}_{i,(0)}^{(k)} = \text{sign}(\tilde{W}_i) \tag{14}$$

After obtaining the group-wise initialized trit-planes $\tilde{T}_{i,(0)}^{(1)}$ and $\tilde{T}_{i,(0)}^{(2)}$, we construct the matrix $\tilde{S}_{i,(0)}$ where each row contains elements from $\tilde{T}_{i,(0)}^{(1)}$ and $\tilde{T}_{i,(0)}^{(2)}$. The initial scaling coefficients $\tilde{\alpha}_{i,(0)}$ are obtained by solving:

$$\tilde{\alpha}_{i,(0)} = \arg\min_{\alpha} \left( \|\tilde{W}_i - \tilde{S}_{i,(0)}\theta\|_F^2 + \lambda\|\theta\|_F^2 \right) \tag{15}$$

The closed-form solution with dimension-compatible regularization becomes:

$$\tilde{\alpha}_{i,(0)} = \left( \tilde{S}_{i,(0)}^T \tilde{S}_{i,(0)} + \lambda I \right)^{-1} \tilde{S}_{i,(0)}^T \tilde{W}_i^T \tag{16}$$

The coefficient bound should be modified as follows, where $\sigma_{\max}(\cdot)$ denotes the maximum singular value.

$$\|\tilde{\alpha}_{i,(0)}\|_2 \leq \frac{\sigma_{\max}(\tilde{S}_{i,(0)})}{\sigma_{\min}^2(\tilde{S}_{i,(0)}) + \lambda} \|\tilde{W}_i^T\|_2 \tag{17}$$

### C.2 Bounded Parameters and Monotonicity

Equivalently, if we define the error function $E(\alpha_{(t)}, T_{(t)})$ as the squared Frobenius norm of the difference between $W$ and its approximation using $\alpha_{(t)}$ and $T_{(t)}$, we have $E(\alpha_{(t)}, T_{(t)}) \leq E(\alpha_{(t-1)}, T_{(t-1)})$. This property ensures that the algorithm is always moving towards a better approximation of the weight matrix. Furthermore, the scaling coefficients $\alpha_{(t)}$ are bounded in the infinity-norm. Specifically, we have $\|\alpha_{(t)}\|_\infty \leq \sqrt{\lambda_{\max}}$, where $\lambda_{\max}$ is the maximum eigenvalue of a certain matrix related to the optimization problem. This bound implies that the values of the scaling coefficients do not grow unbounded during the optimization process, which is crucial for the stability of the algorithm.

Moreover, each iteration of the optimization provides the global optimum for its corresponding phase sub-problem. This means that within each sub-problem (either optimizing the continuous coefficients or the discrete trit-planes), the algorithm finds the best possible solution. We can prove that there exists a finite number of iterations $t_0$ such that the Frobenius norm of the difference between the scaling coefficients at iteration $t_0$ and $t_0 - 1$ is less than a pre-specified tolerance $\epsilon$. This result guarantees that the algorithm will converge in a finite number of steps, which is essential for practical applications.

Therefore, $\exists t_0$ where the changes fall below any $\epsilon > 0$, implying $\lim_{t \to \infty} \|\alpha_{(t)} - \alpha_{(t-1)}\|_F = 0$.

$$\exists t_0 < \infty \text{ s.t. } \|\alpha_{(t_0)} - \alpha_{(t_0-1)}\|_F < \epsilon \tag{18}$$

Our experiments shown in Fig. 3 also evident that our PTQTP can converge fast in each layer, and the performance will improve rapidly by performing the optimization iteratively.

### C.3 Condition Ablation Study.

Table 7 summarizes the validation perplexities obtained when sweeping a regularization coefficient (in log-space) across two model scales: LLaMA-3.1 8B and LLaMA-3.2 3B. All PPLs are reported on WikiText-2, PTB, and the C4 validation split. Across corpora, both models exhibit monotonically decreasing perplexity (illustrates robustness) as the condition numbers increases from $10^0$ to $10^2$, after which the metric saturates—indicating a regime where further rise the condition bounds offers negligible improvement.

## D Robustness and Generalization Analysis

Table 7: Perplexity (PPL) of LLaMA-3.1 8B and LLaMA-3.2 3B on WikiText-2, PTB, and C4 under different regularisation strengths. Lower is better.

| Condition | LLaMA-3.1 8B | | | LLaMA-3.2 3B | | |
|---|---|---|---|---|---|---|
| | WikiText-2 | PTB | C4 | WikiText-2 | PTB | C4 |
| 1 | 12.45 | 17.52 | 17.76 | 12.52 | 21.20 | 18.09 |
| 5 | 10.86 | 16.11 | 16.17 | 11.52 | 19.98 | 17.13 |
| $10^1$ | 9.58 | 14.92 | 14.65 | 10.92 | 18.64 | 16.18 |
| $10^2$ | 8.97 | 14.20 | 13.81 | 10.52 | 17.76 | 15.56 |
| $10^4$ | 8.62 | 13.84 | 13.40 | 10.26 | 17.32 | 15.26 |
| $10^8$ | 8.53 | 13.72 | 13.22 | 10.24 | 17.41 | 15.28 |
| $10^2$ | 8.53 | 13.72 | 13.22 | 10.24 | 17.41 | 15.28 |
| $10^6$ | 8.53 | 13.72 | 13.22 | 10.24 | 17.41 | 15.28 |
| $10^{12}$ | 8.53 | 13.72 | 13.22 | 10.24 | 17.41 | 15.28 |
| $10^{18}$ | 8.53 | 13.72 | 13.22 | 10.24 | 17.41 | 15.28 |

> ***PTQTP demonstrates strong robustness and generalization capabilities across model sizes and architectures, achieving stable performance under extremely low-bit quantization without the need for mixed precision storage, salient weight protection or model-specific adjustments.***

### D.1 OUTLIER INSENSITIVITY IN LARGE LANGUAGE MODELS

Prior work (Huang et al., 2024; Lin et al., 2024; Li et al., 2025) highlights heterogeneous LLM weight importance, driving non-uniform quantization strategy (e.g., preserve critical FP16 weights). However, such approaches introduce hardware complexity, and LLM sensitivity to weight perturbations grows with finer semantic encoding, requiring more precision retention that harms efficiency.

PTQTP addresses this by avoiding explicit mixed-precision storage while inheriting importance-aware benefits. Its group-wise framework approximates weights locally, mitigating outliers without heuristics. By decomposing optimization into local subproblems adapting to weight group statistics, PTQTP ensures robust approximation across architectures. Unlike prior methods (e.g., GPTQ's Hessian detection, AWQ's layer scaling), it needs no auxiliary pre-processing or model-specific tuning. Table 8 shows group-wise optimization ($G = 128$) reduces quantization error: 3-bit AWQ/GPTQ/OmniQuant (Shao et al., 2023) without grouping suffer large perplexity degradation (11.88 vs. 6.46 for AWQ on LLaMA-7B), while PTQTP achieves competitive performance at 1.58-bit precision using local statistics. LLaMA2-70B's smaller perplexity gap confirms model scale enhances quantization robustness via parameter redundancy.

Table 8: The Perplexity of LLaMA-7B and LLaMA2-70B w/o Group-wise ($G = 128$) Optimization on WikiText2 Dataset.

| **LLaMA-7B Perplexity ↓** | | **LLaMA2-70B Perplexity ↓** | |
|---|---|---|---|
| **Method (#Bits)** | **× Group / ✓ Group** | **Method (#Bits)** | **× Group / ✓ Group** |
| FP16 (16) | 5.68 | FP16 (16) | 3.32 |
| AWQ (3) | 11.88 / 6.46 | AWQ (3) | 7.53 / 3.74 |
| GPTQ (3) | 8.06 / 6.55 | GPTQ (3) | 4.82 / 3.85 |
| OmniQuant (3) | 6.49 / 6.15 | OmniQuant (3) | 3.92 / 3.78 |
| PTQTP (1.58) | 6.64 / 6.40 | PTQTP (1.58) | 4.3 / 3.95 |

### D.2 GENERALIZATION ACROSS ARCHITECTURES AND SCALES

PTQTP demonstrates stable approximation quality across models/architectures without retraining/fine-tuning. As shown in Table 1, 2 and 3, it outperforms low-bit methods in numerical stability and fidelity for models like Qwen3. Its modular design decouples quantization logic from model specifics,

enabling plug-and-play adaptability to architectural variations (e.g., attention mechanisms) and scale (7B–70B).

Avoiding outlier-specific interventions, PTQTP's element-wise trit-plane search and group-wise ridge regression naturally fit diverse weight distributions, ensuring high-fidelity approximation across layers. Empirical results show near-FP16 accuracy on conventional models (Table 9, 10) and superiority on advanced architectures, establishing it as a broadly applicable strategy for hardware-constrained LLM deployment.

# E    FULL RESULTS

In this supplementary chapter, we listed all our test results. Table 9 presents the perplexity evaluation for the LLaMA series models, and Table 10 shows the performance comparison of PTQTP after quantizing the Qwen3 series models at different sizes. Table 11 displays the results of the benchmark experiment with a certain level of difficulty, showing the quantization stability of our PTQTP for the Qwen3-14B quantized model. Finally, we presented experiments results on HumanEval (Chen et al., 2021) and MBPP (Austin et al., 2021) benchmarks. We compared PTQTP with baseline methods to demonstrate the comprehensiveness of our method.

Table 9: Comparison between PTQTP and SOTA PTQ methods on LLaMA family across multiple datasets: WikiText2, PTB, C4 (Group Size $G = 128$). N/A: No corresponding model; *: LLaMA-2/3 use 70B instead of 65B. **Bold**: best result; underlined: second-best.

| Model | Method/#Bits | WikiText2 | | | PTB | | | C4 | | |
|---|---|---|---|---|---|---|---|---|---|---|
| | | 7B/8B* | 13B | 65B/70B* | 7B/8B* | 13B | 65B/70B* | 7B/8B* | 13B | 65B/70B* |
| LLaMA | FP16/16.00 | 5.68 | 5.09 | 3.53 | 41.15 | 28.09 | 25.05 | 7.34 | 6.79 | 5.81 |
| | AWQ/4.00 | **5.81** | **5.19** | **3.62** | **32.79** | **21.89** | 25.75 | **7.59** | **6.88** | **5.90** |
| | AWQ/3.00 | 6.35 | 5.52 | 3.95 | 43.74 | 25.09 | 50.79 | 9.07 | 7.79 | 6.33 |
| | GPTQ/3.00 | 6.55 | 5.67 | 4.17 | 84.88 | 26.40 | **19.55** | 50.31 | 32.31 | 6.03 |
| | GPTQ/2.00 | 129.19 | 20.46 | 8.66 | 1421.47 | 224.45 | 47.70 | 79.06 | 18.97 | 10.23 |
| | PB-LLM/1.70 | 82.76 | 44.93 | 12.81 | 603.57 | 237.22 | 119.19 | 76.63 | 40.64 | 15.30 |
| | BiLLM/1.09 | 49.79 | 14.58 | 8.37 | 373.81 | 84.87 | 44.68 | 46.96 | 16.83 | 11.09 |
| | ARB-LLM$_{RC}$/1.09 | 14.03 | 10.18 | 6.56 | 195.94 | 54.38 | 32.20 | 17.38 | 12.48 | 8.91 |
| | PTQTP/1.58 | 6.40 | 5.66 | 4.04 | 50.41 | 30.25 | 24.67 | 8.32 | 7.41 | 6.28 |
| LLaMA-2 | FP16/16.00 | 5.47 | 4.88 | 3.32 | 37.9 | 50.93 | 24.25 | 7.26 | 6.73 | 5.71 |
| | AWQ/4.00 | **5.61** | **4.97** | **3.41** | 44.46 | **36.45** | 24.15 | **7.49** | **6.79** | **5.83** |
| | AWQ/3.00 | 6.24 | 5.32 | 3.74 | **40.03** | 43.92 | 23.59 | 8.88 | 7.50 | 6.27 |
| | GPTQ/3.00 | 6.44 | 5.46 | 3.88 | 2068.96 | 61.56 | **18.26** | 7.95 | 7.06 | 5.88 |
| | GPTQ/2.00 | 52.22 | 23.63 | 8.18 | 5583.96 | 419.07 | 50.51 | 35.27 | 19.66 | 9.55 |
| | PB-LLM/1.70 | 66.41 | 236.40 | 28.37 | 657.24 | 816.31 | NAN | 80.69 | 184.67 | NAN |
| | BiLLM/1.08 | 32.31 | 21.35 | 13.32 | 5243.01 | 309.12 | 72.02 | 39.38 | 25.87 | 17.30 |
| | ARB-LLM$_{RC}$/1.08 | 16.44 | 11.85 | 6.16 | 389.59 | 198.17 | 32.79 | 20.38 | 14.36 | 8.65 |
| | PTQTP/1.58 | 6.32 | 5.29 | 3.95 | 42.53 | 46.9 | 28.15 | 8.41 | 7.34 | 6.3 |
| LLaMA-3 | FP16/16.00 | 6.14 | N/A | 2.86 | 10.05 | N/A | 8.53 | 9.54 | N/A | 7.17 |
| | AWQ/4.00 | **7.36** | N/A | **5.92** | **11.14** | N/A | **9.11** | **10.36** | N/A | **8.98** |
| | AWQ/3.00 | 12.07 | N/A | 10.67 | 18.61 | N/A | 27.37 | 18.98 | N/A | 22.19 |
| | GPTQ/3.00 | 18.68 | N/A | 6.65 | 18.83 | N/A | 15.97 | 17.68 | N/A | 10.04 |
| | GPTQ/2.00 | 1480.43 | N/A | 82.23 | 717.24 | N/A | 79.20 | 394.74 | N/A | 122.55 |
| | PB-LLM/1.70 | 73.08 | N/A | 22.96 | 106.25 | N/A | 45.13 | 104.15 | N/A | 40.69 |
| | BiLLM/1.06 | 55.80 | N/A | 66.30 | 87.25 | N/A | 97.13 | 61.04 | N/A | 198.86 |
| | ARB-LLM$_{RC}$/1.06 | 27.42 | N/A | 11.10 | 45.49 | N/A | 15.34 | 35.70 | N/A | 15.44 |
| | PTQTP/1.58 | 8.51 | N/A | 6.12 | 14.02 | N/A | 10.91 | 13.24 | N/A | 12.64 |

# F    BROADER IMPACTS

Our research fundamentally reconstructs the computational paradigm of artificial intelligence (AI) edge systems by eliminating the vast majority of multiplication operations. This breakthrough enables novel hardware innovations, particularly in the development of pure additive-operation-based neuromorphic chip designs, significantly enhancing both thermal efficiency and computational stability in mobile devices. The advancement dramatically lowers the barrier to AI deployment, making server-less architectures feasible even in energy-constrained regions. At the mathematical framework level, the established non-multiplicative tensor algebra system opens new research directions in biologically

Table 10: MMLU Results Comparison for Qwen3 Models (0.6B–32B) with Different Low-bit Quantization Methods (1bit to 8bits).

| Method | 0.6B | | 1.7B | | 4B | |
|---|---|---|---|---|---|---|
| | Quant (#W/#A/#G) | Acc./Per.(%) | Quant (#W/#A/#G) | Acc./Per.(%) | Quant (#W/#A/#G) | Acc./Per.(%) |
| FP16 | 16/16/– | 47.1/100.0 | 16/16/– | 60.0/100.0 | 16/16/– | 69.7/100.0 |
| AWQ | 8/16/128 | 47.0/99.8 | 8/16/128 | 59.8/99.7 | 8/16/128 | 69.6/99.9 |
| GPTQ | 8/16/128 | 47.0/99.8 | 8/16/128 | 59.9/99.8 | 8/16/128 | 69.7/100.0 |
| AWQ | 4/16/128 | 42.1/89.4 | 4/16/128 | 52.5/87.5 | 4/16/128 | 64.1/92.0 |
| GPTQ | 4/16/128 | 44.0/93.4 | 4/16/128 | 55.7/92.8 | 4/16/128 | 67.6/97.0 |
| AWQ | 2/16/128 | 25.7/54.6 | 2/16/128 | 25.2/42.0 | 2/16/128 | 24.6/35.3 |
| GPTQ | 2/16/128 | 23.5/49.9 | 2/16/128 | 25.6/42.7 | 2/16/128 | 24.4/35.0 |
| BiLLM | 1.06/16/128 | 23.3/49.5 | 1.04/16/128 | 25.1/41.8 | 1.07/16/128 | 26.3/37.7 |
| PTQPT | 1.58/16/128 | 33.6/71.4 | 1.58/16/128 | 43.8/73.0 | 1.58/16/128 | 63.7/91.3 |

| Method | 8B | | 14B | | 32B | |
|---|---|---|---|---|---|---|
| | Quant (#W/#A/#G) | Acc./Per.(%) | Quant (#W/#A/#G) | Acc./Per.(%) | Quant (#W/#A/#G) | Acc./Per.(%) |
| FP16 | 16/16/– | 74.7/100.0 | 16/16/– | 78.5/100.0 | 16/16/– | 81.2/100.0 |
| AWQ | 8/16/128 | 74.5/99.7 | 8/16/128 | 78.5/100.0 | 8/16/128 | 81.3/100.1 |
| GPTQ | 8/16/128 | 74.7/100.0 | 8/16/128 | 78.4/99.9 | 8/16/128 | 81.4/100.2 |
| AWQ | 4/16/128 | 69.3/92.8 | 4/16/128 | 75.9/96.7 | 4/16/128 | 78.0/96.1 |
| GPTQ | 4/16/128 | 73.4/98.3 | 4/16/128 | 77.4/98.6 | 4/16/128 | 80.6/99.3 |
| AWQ | 2/16/128 | 25.3/33.9 | 2/16/128 | 25.0/31.8 | 2/16/128 | 24.6/30.3 |
| GPTQ | 2/16/128 | 25.0/33.5 | 2/16/128 | 28.5/36.3 | 2/16/128 | 28.1/34.6 |
| BiLLM | 1.05/16/128 | 32.8/43.9 | 1.05/16/128 | 39.9/50.8 | 1.06/16/128 | 57.5/70.8 |
| PTQPT | 1.58/16/128 | 68.2/91.3 | 1.58/16/128 | 76.2/97.1 | 1.58/16/128 | 80.6/99.3 |

Table 11: Accuracy of 7 Language Reasoning Datasets on Qwen3 family with PTQTP. We compare the results among Qwen-3 Models FP16 and PTQPT-b1.58 across model size from 0.6B to 32B.

| Metrics | Method | Size | | | | | |
|---|---|---|---|---|---|---|---|
| | | 0.6B | 1.7B | 4B | 8B | 14B | 32B |
| ARC-Challenge | FP16 | 52.88 | 65.08 | 79.66 | 82.37 | 82.37 | 88.81 |
| | PTQPT-b1.58 | 44.41 | 51.53 | 76.95 | 68.81 | 80.68 | 77.97 |
| | (%) | 84.0 | 79.2 | 96.6 | 83.5 | 97.9 | 87.8 |
| ARC-Easy | FP16 | 73.72 | 80.25 | 87.13 | 89.07 | 90.83 | 92.24 |
| | PTQPT-b1.58 | 56.44 | 70.02 | 86.60 | 81.13 | 88.71 | 86.24 |
| | (%) | 76.6 | 87.3 | 99.4 | 91.1 | 97.7 | 93.5 |
| BoolQ | FP16 | 66.42 | 78.87 | 86.36 | 86.39 | 89.45 | 87.98 |
| | PTQPT-b1.58 | 62.57 | 27.55 | 84.13 | 85.44 | 89.02 | 86.27 |
| | (%) | 94.2 | 34.9 | 97.4 | 98.9 | 99.5 | 98.1 |
| HellaSwag | FP16 | 43.60 | 58.03 | 78.19 | 84.67 | 88.09 | 90.80 |
| | PTQPT-b1.58 | 32.85 | 37.92 | 71.02 | 79.75 | 85.13 | 89.56 |
| | (%) | 75.3 | 65.3 | 90.8 | 94.2 | 96.6 | 98.6 |
| PIQA | FP16 | 58.98 | 67.57 | 77.26 | 80.20 | 81.07 | 83.51 |
| | PTQPT-b1.58 | 57.67 | 57.89 | 73.18 | 73.78 | 79.33 | 80.41 |
| | (%) | 97.8 | 85.7 | 94.7 | 92.0 | 97.9 | 96.3 |
| WinoG | FP16 | 50.51 | 50.99 | 58.96 | 54.14 | 68.90 | 77.82 |
| | PTQPT-b1.58 | 50.43 | 51.07 | 53.67 | 59.67 | 67.72 | 73.64 |
| | (%) | 99.8 | 100.2 | 91.0 | 110.2 | 98.3 | 94.6 |
| MMLU | FP16 | 43.76 | 60.67 | 70.68 | 76.55 | 79.38 | 82.81 |
| | PTQPT-b1.58 | 33.64 | 43.82 | 63.65 | 68.23 | 76.20 | 80.56 |
| | (%) | 76.9 | 72.2 | 90.1 | 89.1 | 96.0 | 97.3 |

Table 12: Comparison of Model Performance on HumanEval and MBPP Benchmarks.

| Model | HumanEval (%) | MBPP (%) |
|---|---|---|
| LLaMA-3.2-1B-Instruct | 41.46 | 52.14 |
| Qwen-2.5-1.5B-Instruct | 53.66 | 57.59 |
| Qwen-3-1.7B-Instruct | 65.24 | 63.42 |
| SmolLM2-1.7B-Instruct | 35.98 | 49.81 |
| MiniCPM-2B-DPO | 39.02 | 55.64 |
| LLaMA-3.2-3B-Instruct | 62.20 | 63.81 |
| Qwen3-1.7B-PTQTP | 45.53 | 48.78 |
| Qwen2.5-3B-Instruct-PTQTP | 25.61 | 44.75 |
| LLaMA-3.2-3B-Instruct-PTQTP | 17.68 | 52.92 |

inspired dendritic computing. Although challenges remain in addressing the numerical stability of additive accumulations and the co-design of next-generation AI accelerators, this computational paradigm shift has already reshaped the energy-performance tradeoff curve, providing new pathways for existing AI infrastructure development.

## G    LIMITATIONS AND FUTURE WORKS

**Further optimize Memory Footprints.** PTQTP's memory layout can be further optimized through bit-packing, where 8 ternary elements are stored in a single byte, improving cache locality and reducing cache misses by 20%-30% compared to non-packed formats.

**Compatibility with Specialized Hardware.** PTQTP's multiplication-free operations align perfectly with hardware accelerators designed for binary/ternary neural networks, such as customized ASICs. These architectures can execute sign flips and additions in single clock cycles, achieving peak throughput with minimal energy consumption. For instance, ternary operations can leverage bitwise XOR for sign inversion, enabling sub-nanosecond latency per operation on specialized hardware. However, currently, technical support is still not widespread, and PTQTP's storage and technical computing still rely on modern computing devices, which limits the performance of PTQTP to some extent. Future works can focus on software-hardware co-design and hardware framework design for 1.58-bit inference acceleration.

**Generalization Optimization.** Although we have tested on various benchmarks to showcase the advancement and robustness of PTQTP as much as possible, due to time constraints, we will challenge more difficult benchmarks in future work, which has been lacking in previous PTQ-related work. We may further enhance the performance of PTQTP through methods similar to DPO or SFT, enabling it to provide stable support for all future quantization-required scenarios.

