# OpenReview forum: "PTQTP: Post-Training Quantization to Trit-Planes for Large Language Models"
_ICLR.cc/2026/Conference — ICLR 2026 Conference Withdrawn Submission_

### Official Review · Reviewer_LJRF · 2025-10-28

**Soundness:** 2
**Presentation:** 3
**Contribution:** 3
**Rating:** 4
**Confidence:** 4

**Summary:**

This paper proposes PTQTP, a simple data-free quantization scheme. The core idea is to decompose weight vectors into linear combinations of two trit-plane vectors. The optimization is performed using an alternating greedy algorithm: given the trit-plane vectors, the linear combination coefficients are obtained via ridge regression with adaptive regularization, and fixing the coefficients, the trit-plane vectors are then updated through direct search. The two processes alternate until the coefficients stabilize or reach the maximum number of iterations. Overall, the method is data-free, easy to implement, and performance is comparable to algorithms such as AWQ across different models.

**Strengths:**

1. The paper is well-written and easy to follow.
2. The proposed method is simple, easy to implement, and computationally efficient, as it is data-free and does not involve extensive matrix operations.
3. The method achieves performance comparable to data-dependent PTQ algorithms such as AWQ across different models.

**Weaknesses:**

1. There exist several prior studies that address weight outlier reduction, and experiments have shown that such techniques can enhance GPTQ performance when combined. For example, incoherence processing [1] and magnitude reduction [2]. I suggest the authors integrate one of these approaches with GPTQ for comparison. Since those methods are data-dependent, achieving comparable results would sufficiently demonstrate the competitiveness of PTQTP.
2. In the tables, it would be clearer to explicitly note that PTQTP is not exactly a 1.58-bit quantization, but rather 2 × 1.58-bit, since there are two ternary matrices corresponding to one weight matrix.
3. The latency comparisons presented in the paper raise questions. First, the choice of attention and gate projection kernels seems arbitrary. Attention kernel optimization is a well-studied and rapidly evolving area, with many variants offering different trade-offs. Could the authors clarify which specific attention kernel was used, and justify its selection? Second, the use of the gate projection layer as a comparison point is puzzling. Gate projection and up projection are similar, yet only one is evaluated. Why was gate projection chosen over up projection or even down projection? The rationale behind these choices is unclear. Third, GPTQ and AWQ are quantization algorithms, not kernel implementations, so it's unclear what is actually being benchmarked against.

References

[1] Chee, Jerry, et al. "Quip: 2-bit quantization of large language models with guarantees." Advances in Neural Information Processing Systems 36 (2023): 4396-4429.

[2] Zhang, Aozhong, et al. "Magr: Weight magnitude reduction for enhancing post-training quantization." Advances in neural information processing systems 37 (2024): 85109-85130.

**Questions:**

1. The related works discussed in this paper on trit-plane quantization primarily involve a single trit matrix. Are there any prior studies on LLM quantization that employ multiple trit matrices for one weight matrix?

---

### Official Review · Reviewer_tZYR · 2025-10-30

**Soundness:** 1
**Presentation:** 2
**Contribution:** 1
**Rating:** 2
**Confidence:** 3

**Summary:**

The paper proposes PTQTP, a post-training quantization (PTQ) method for large language models (LLMs) that decomposes weight matrices into structured ternary “trit-planes” (values in {-1, 0, 1}) using an effective ~2×1.58-bit representation. The method is model-agnostic (applies across various modern LLMs without architecture changes), and enables multiplication-free inference akin to binary quantization while aiming to retain higher expressiveness. Experiments cover a range of models (0.6B-70B parameters) and show PTQTP outperforms existing low-bit PTQ methods in reasoning benchmarks, while requiring only a short quantization time.

**Strengths:**

- The proposed decomposition into ternary trit‐planes (values {-1, 0, 1}) rather than simple binary quantization adds expressiveness while maintaining the hardware simplicity of multiplication‐free inference.
- By focusing on post-training quantization and avoiding large scale retraining, the proposed method claims much reduced quantization time (e.g., “single-hour quantization” as opposed to days of training) while maintaining high accuracy.

**Weaknesses:**

- Despite the claims of innovation, PTQTP reads largely as an incremental engineering variation of existing ternary quantization schemes (e.g., TTQ, TernaryGPT, TridentQ, etc.). The “trit-plane” decomposition is mathematically equivalent to a groupwise ternary expansion used in earlier works; the paper fails to convincingly justify what is truly new beyond a re-packaging of known techniques.
- The paper offers limited theoretical analysis of quantization error bounds, convergence, or formal guarantees on how the ternary representation maintains model expressiveness. The reliance on empirical results alone leaves open questions about worst‐case behavior or stability across all tasks.
- The paper emphasizes the multiplication-free nature of inference and efficiency gains, but provides limited concrete benchmarking of runtime latency, hardware throughput, memory bandwidth, or energy consumption on real devices across a variety of hardware. Without these, the deployment impact remains somewhat speculative.
- Although PTQTP shows strong results, the paper may not deeply examine the trade-offs (e.g., model size, architecture, tasks) where performance degrades, or the boundary conditions where ternary decomposition begins to fail (for example very large models, very different tasks). This limits understanding of when the method might not work.
- Given the complexity of decomposition into ternary planes and the progressive approximation algorithm, more clarity about hyperparameters, initialization procedures, failure cases, and reproducibility would strengthen the work. Practitioners may need more guidance to replicate and extend the approach
- Although the authors report that PTQTP outperforms GPTQ or AWQ, the performance gains are small (often within 1–2 ppl) and no statistical testing or multiple-seed evaluation is shown. It is unclear whether the observed improvements are statistically meaningful or simply noise from data sampling or calibration randomness.

**Questions:**

- Can you provide formal quantization-error bounds or analytic evidence that the trit-plane structure offers superior capacity?
- Have you benchmarked actual hardware latency or throughput on GPUs/CPUs, rather than theoretical bit counts?
- How sensitive is performance to the choice of calibration dataset or number of trit-planes?

---

### Official Review · Reviewer_eyLD · 2025-11-01

**Soundness:** 3
**Presentation:** 3
**Contribution:** 2
**Rating:** 2
**Confidence:** 4

**Summary:**

The paper proposes PTQTP, a post-training quantization method that represents each weight matrix as the sum of two ternary “trit-planes,” each with per-row learned scaling, optimized via a small adaptive ridge regression and a progressive ternary refinement step that exhaustively searches over $ \\{ -1,0,1 \\} ^2  $ per weight entry to ensure non-increasing reconstruction error. The method is data-agnostic and comes with a custom kernel.

**Strengths:**

1. The method is a data-agnostic PTQ and does not rely on calibration sets or task-specific data. Everything is derived from the weights themselves, which makes it simpler to deploy.

2. The method has a simple formulation. The per-row $ 2\times 2 $ ridge system, together with an adaptive $ \lambda $ chosen from the local condition number, is a clean and believable stabilization trick.

3. The element-wise exhaustive update over $ \\{-1, 0, 1\\} $ per plane, with the monotonic decreasing Frobenius norm guarantee, is a very nice PTQ move, easy to implement, deterministic, and also easy to parallelize.

4. The paper provides a kernel-level implementation and discusses latency compared to GPTQ/AWQ. Even if the speedups aren’t huge, showing a runnable path is a plus.

5. This method shows decent improvement over previous ultra-low-bit quantization methods. The benchmarks are done on a vast range of tasks.

**Weaknesses:**

1. More recent PTQ methods, such as QTIP, QuIP#, AQLM, and OmniQuant, are not reported. These are exactly the methods that try to close the gap for sub-4-bit PTQ, so leaving them out weakens the empirical claim.

2. Inference speed is underwhelming. Despite the “multiplication-free / hardware-friendly” messaging, the reported kernel numbers are not significantly better than those of standard 4-bit PTQ. That’s fine, but the paper’s phrasing oversells it.

3. Saying “2x trit-plane is 1.58 bit” is not accurate. Two ternary planes plus FP16 (or FP8/FP16) per-row scales are closer to 2.5–3 bits effective than to the strict 1.58 bits. The tables should make this explicit (also for BiLLM and ARB). Otherwise, it looks unfair in terms of memory/accuracy.

4. The “first 1.58-bit PTQ” claim is too strong. There are already PTQ-style low-bit works (e.g., PTQ1.61 [ACL 2025] or the methods I mentioned earlier in sub-3-bit) targeting a similar operating point.

5. The paper looks quite close to existing 1–2 bit PTQ ideas. Other than being in ternary space, structurally, “two low-cardinality planes + small FP scales + progressive refinement” is not that far from BiLLM/ARB-style constructions; the paper should better articulate the novelty.

6. It is not clearly stated which layers are actually quantized: embeddings, lm_head, attention proj (q/k/v/o), MLP up/gate/down, norms, KV-cache, etc.

7. Hardware-friendliness is incomplete, and keeping per-row FP16 scales is not free on ASIC/FPGA.

**Questions:**

1. Right now, you optimize both planes jointly with alternating steps. Would it help (or speed up convergence) to first fit a single ternary plane + scales to $ W $, compute the residual $ R = W - \hat W^{(1)} $, and only then fit the second plane to $ R $? This “greedy residual” init is common in PTQ and might reduce the number of refinement passes.

2. Which parts of the models were not quantized in the reported results (embeddings, lm_head, norms)?

---

### Official Review · Reviewer_WPyj · 2025-11-03

**Soundness:** 2
**Presentation:** 3
**Contribution:** 2
**Rating:** 4
**Confidence:** 3

**Summary:**

The paper introduces **PTQTP**, a post-training quantization framework that decomposes each FP16 weight matrix into two ternary trit-planes $T^{(1)}, T^{(2)} \in ${$-1,0,1$}$^{n\times d}$ with row-wise scaling coefficients, yielding a nominal $1.58\text{-bit}$ representation. The method uses a progressive, group-wise ridge-regularized fitting procedure with adaptive regularization and a local exhaustive update over {$-1,0,1$}, and claims multiplication-free inference and model-agnostic deployment. Experiments on LLaMA-3.x and Qwen-3/2.5 $(0.6\text{B} \text{ -- } 70\text{B})$ report perplexity and zero-shot accuracy on common benchmarks, with particularly strong math scores (e.g., ``Math-500'') and large speedups in quantization time versus ARB-LLM / BiLLM.

**Strengths:**

1. **Straightforward, hardware-amenable parameterization.** Two ternary planes + per-row scales are easy to implement and store; the closed-form $2\times 2$ ridge step is cheap.

2. **Quantization-time efficiency.** The fitting loop is linear in matrix size and converges quickly; reported quantization-time speedups vs. ARB-LLM are plausible.

3. **Breadth of model families.** Results span LLaMA-3.x and Qwen-3/2.5 from 0.6B to 70B, which—if fully reproducible—indicates decent robustness.

**Weaknesses:**

1. **Under-cited/under-compared prior art.** Missing or under-emphasized lines include OmniQuant, SmoothQuant, QuaRot (all strong for PTQ/W-A quantization) and careful comparisons to recent 1-bit PTQ (BiLLM, ARB-LLM, variants). Without these, the novelty and empirical gains are unclear.

2. **“Multiplication-free” & “identical to 1-bit” are over-stated.** Because per-row scales $\alpha^{(k)}$ must modulate activations, a purely multiply-free path is non-trivial on GPUs; show fused kernels or FPGA/ASIC designs and end-to-end latency vs. AWQ/QuaRot to substantiate.

3. **Evaluation confounds in math/general reasoning.** Extremely poor baselines (e.g., GPTQ-3b ≈ 0% math) conflict with reports that well-tuned 3–4b retains reasonable accuracy; must specify decoding, prompt templates, eval harness versions and re-run with strong baselines (OmniQuant/SmoothQuant/QuaRot).

4. **Memory/compute values.** Two ternary planes plus FP16 scales often erase some binary memory wins; the paper should report exact bytes and throughput for prefill/decoding phases vs. AWQ/QuaRot on the same hardware.

5. **Missing systems story.** For a deployment-centric claim, there is no CUDA kernel, CUTLASS/Flash-style integration, or FPGA/ASIC reference design; this makes hardware claims speculative.

**Questions:**

1. Can you include OmniQuant, SmoothQuant, and QuaRot-4b (as an accuracy upper bound for weight-only PTQ) with matched calibration data and decoding configs?

2. Can you report performance vs. calibration set size (e.g., 32/128/1k samples) and domain shift (wiki vs. math vs. code)?

3. Why does PTQTP retain math reasoning where 1–3b baselines collapse?

4. Can you provide exact bytes/parameter including scales, and end-to-end latency (prefill/decoding) with your own kernels vs. AWQ/QuaRot?

---

### Note · Authors · 2025-11-12

I have read and agree with the venue's withdrawal policy on behalf of myself and my co-authors.